# Long-Term Monitoring of Tree Population Dynamics in Desert Ecosystems: Integrating Field and Satellite Data

Sivan Isaacson [1,*], Jhonathan E. Ephrath [2], Shimon Rachmilevitch [2], Dan G. Blumberg [3,4], Benny Shalmon [5], Ofir Katz [1,6] and Shimrit Maman [4]

1  Dead Sea and Arava Science Center, Mt. Masada 86910, Israel
2  French Associates Institute for Agriculture and Biotechnology of Drylands, Sede Boqer Campus, Ben-Gurion University of the Negev, Beer Sheva 8499000, Israel
3  Geography and Environmental Development, Ben-Gurion University of the Negev, Beer Sheva 8410501, Israel
4  Homeland Security Institute, Ben-Gurion University of the Negev, Beer Sheva 8410501, Israel
5  Israel Nature and Parks Authority, Southern District, Eilat 88100, Israel
6  Eilat Campus, Ben-Gurion University of the Negev, Eilat 8855630, Israel
*  Correspondence: sivanisa@adssc.org

**Abstract:** Arid environments are characterized by rare rain events that are highly variable, as a result of which plant populations often exhibit episodic recruitment and mortality dynamics. However, direct records and observations of such events are rare because of the slow development of woody species. In this study, we described how a decrease in annual precipitation affected acacia tree population dynamics in two hydrological regime types: small wadis and salt flats. This study combines 15 years of continuous, yearly field monitoring of individual acacia trees and data from a historical Corona satellite image, which has extended the time scope of the research. Results indicate that the annual mortality of acacia trees in small wadis reflects the cumulative effective rain events in the preceding five years, whereas the population on the salt flats was not affected by annual rainfall fluctuations. Moreover, in small wadis, rain events of less than 8 mm did not increase acacia tree survival rates. The mortality pattern and dynamics of each plot was unique, suggesting unsynchronized mortality and recruitment episodes on a regional scale. Mortality in all plots was documented both in "old" trees (i.e., recognized in 1968) and "new" trees (not recognized in 1968), but varied highly between plots. More than 50% of the dead trees recorded at the sites had died during the previous dry period (2000–2010). Combining field monitoring and historical satellite image data provided a unique database of acacia population dynamics. This record revealed the response of the acacia population to climate fluctuations and a period of episodic mortality.

**Keywords:** acacia; climate fluctuations; hyper arid; remote sensing; trees episodic mortality

## 1. Introduction

In arid regions, scattered trees such as acacia (genus *Vachellia*) [1] play a significant role in the ecosystem by moderating the extreme environmental conditions [2,3]. These "fertile islands" reduce high radiation and temperature and increase humidity under their canopies [4]. The soil under the canopy is richer in organic matter and nutrients than that outside the canopy [5]; consequently, more plant species and biomass are found near the trees [6,7]. Thus, acacia trees in arid zones are considered to be "keystone species" because they support biodiversity by providing diverse microhabitats and resources to plants and animals [2]. In addition, they are an essential resource for local human populations, providing food for livestock and wood for fuel and construction [8]. In Israel, this phenomenon is rare compared to regions in Africa and the Middle East because tree cutting is illegal [9].

Acacia are long-lived trees that can tolerate several years of drought [10]. Estimations suggest an average potential lifespan of 200 years with older individuals reaching 650 [11].

Therefore, understanding the acacia's response to climate fluctuations is essential when evaluating the vulnerability of the entire ecosystem.

During the last decades, attention has been drawn to reports of high mortality rates and a lack of recruitment in acacia populations of arid and hyper-arid zones of the Middle East (southern Israel [9,12–20], and Qatar [21]) and North Africa (Eastern Desert of Egypt [10,22], Sinai Desert of Egypt [23,24], and pre-Saharan Tunisian desert [25–27]).

Most studies regarding the high mortality and lack of recruitment of acacia populations focus on the impact of anthropogenic disturbance. In Egypt and Qatar, most studies have shown that the main causes of population decline correlate with overgrazing, charcoal production, and wood construction [10,21,23,28]. In Israel, studies suggest that changes in runoff paths due to road and dam construction have led to declining acacia populations [12,14,15,17]. Alteration of runoff paths may also occur naturally during large flash floods and can affect water availability [9,14].

### 1.1. The Geo-Hydrology Environment of Acacia Trees in Israel

Acacia trees in Israel are found where the water supply is adequate and temperatures are high [29]. Because of limited rainfall, the those in southern Israel are usually restricted to ephemeral stream beds (wadis), which are characterized by a higher soil moisture content [3,30]. Trees in large wadis are larger and have higher growth rates compared with trees in small runnels [13]. Yet, no correlation was found between tree mortality and the drainage basin's size or the tree's location within the wadi (i.e., microhabitat) [16]. Alterations of the runoff path (either natural or artificial) were found to correlate strongly with tree mortality and deficient recruitment [9,12,14,15,17]. The effect of aquifer depletion on tree populations was tested but found not to be a significant cause of acacia mortality [17]. In agreement with this claim, Sher et al. [31] used isotopic analysis to show that aquifers are not the source of water found in acacia trees but rather surface water. The Negev is the northern distribution border of the acacia species *A. raddiana* and *A. tortilis* [29], and therefore their morphological and physiological behavior may differ from their origin in the Sudanian phytogeography zone.

### 1.2. Monitoring Vegetation in Arid Regions

In recent decades, threats of desertification and environmental changes have attracted academic attention. Remote sensing of vegetation monitoring is an essential tool and source of information for investigating climate change and desertification impacts on a large scale [32]. Remote sensing archives provide a historical record, making the images an invaluable data source for analyzing changes and trends in vegetation cover over the globe [32]. Even so, remote sensing studies in arid regions have been criticized for focusing mainly on biomass density changes rather than more detailed changes in biodiversity or plant population dynamics [33]. Nevertheless, several studies on scattered tree population dynamics used a remote sensing archive to extract data on individual trees [34–36], but none has compared this data with long-term field monitoring. The objective of this study is to describe how a decrease in annual precipitation affected acacia tree population dynamics in two hydrological regime types: small wadis and salt flats. This study presents observations of individual-based tree population changes by integrating between long-term field monitoring and historical satellite image data. The integration of data sources provides a detailed yet broad picture of acacia population dynamics and their response to the rain and hydrological regime.

## 2. Materials and Methods

### 2.1. Study Area

The study area was in the southern Arava Rift Valley and southern Negev Desert, Israel (Figure 1). The mean accumulated annual rainfall for the area is 30 mm but varies significantly both by time and area [37]. Rain events in this hyper-arid region are rare, and flash floods occur once every few years. Infrequent, high-intensity floods characterize the

area. An apparent decrease in the accumulated annual precipitation in this area has been recorded since 1995 [38] (Figure 2). This decline has reduced the number of flash flood events to nearly zero; consequently, water availability to perennial plants along the wadis has decreased. This sequence of dry years was broken by a regional flood in January 2010 recorded at the area's hydrological stations [16]. The rainy winter of 2010 was followed by more storm events in the following years (Figure 2).

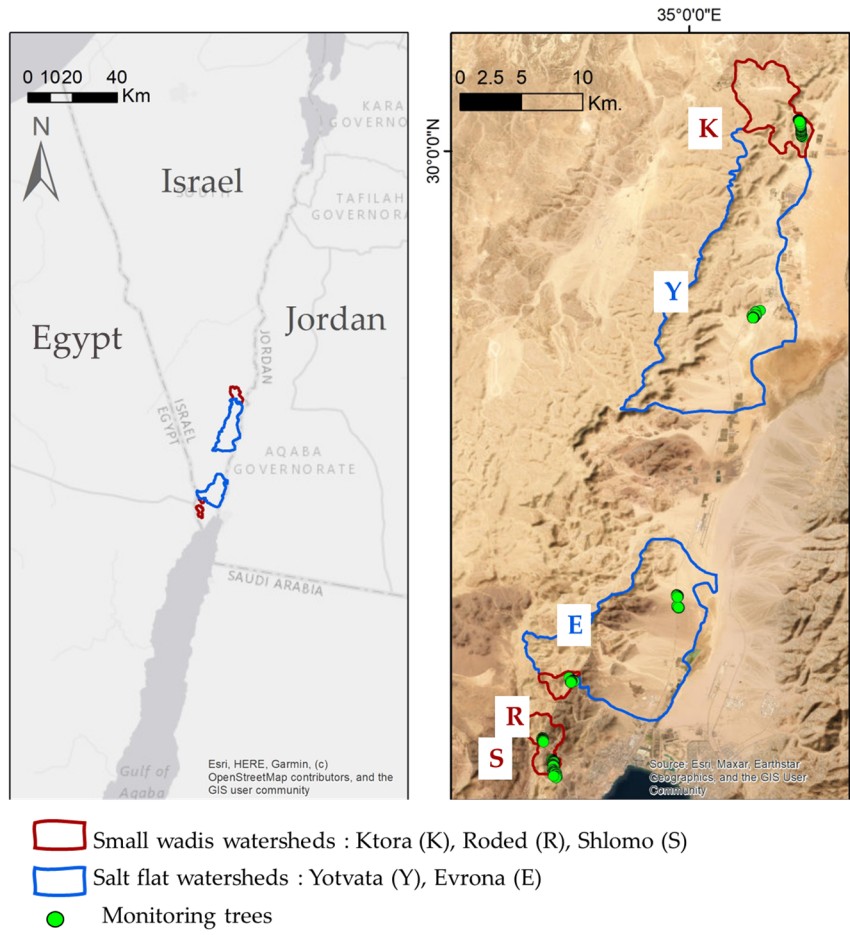

Small wadis watersheds : Ktora (K), Roded (R), Shlomo (S)

Salt flat watersheds : Yotvata (Y), Evrona (E)

Monitoring trees

**Figure 1.** Study area sites and the drainage basin extent for each of the sites.

The study focuses on five sites in the southern Arava Rift Valley. The sites represent two different ecosystems and hydrological regimes: wadis of small catchments (Shlomo, Roded, and Ktora), and the margins of salt flats (Evrona Sabkha, Yotvata Sabkha) (Figure 1).

### 2.1.1. Small Catchment Wadis

The wadis of small catchments selected for this research spread from south to north; the overall average slopes of the flow channels range between 3.6 (Roded) and 7% (Ktora). The two southernmost sites are Wadi Roded and Wadi Shlomo; the lithology of the area is mainly Precambrian magmatic rock. The wadi beds of Roded and Shlomo are composed of alluvium with a mixture of marine carbonates, magmatic rocks, and sandstone. Wadi Ktora, located 50 km north of the area, is incised in marine carbonate alluvium. All the wadis of the small catchment generally flow southeast.

Wadi Shlomo—The wadi flows from northwest to southeast and drains into the Red Sea. Up to the monitoring plot, the wadi drains an area of 10 km². The active channel is incised in magmatic and metamorphic rock; its width ranges between 25 and 80 m in the studied plots (Figure 3).

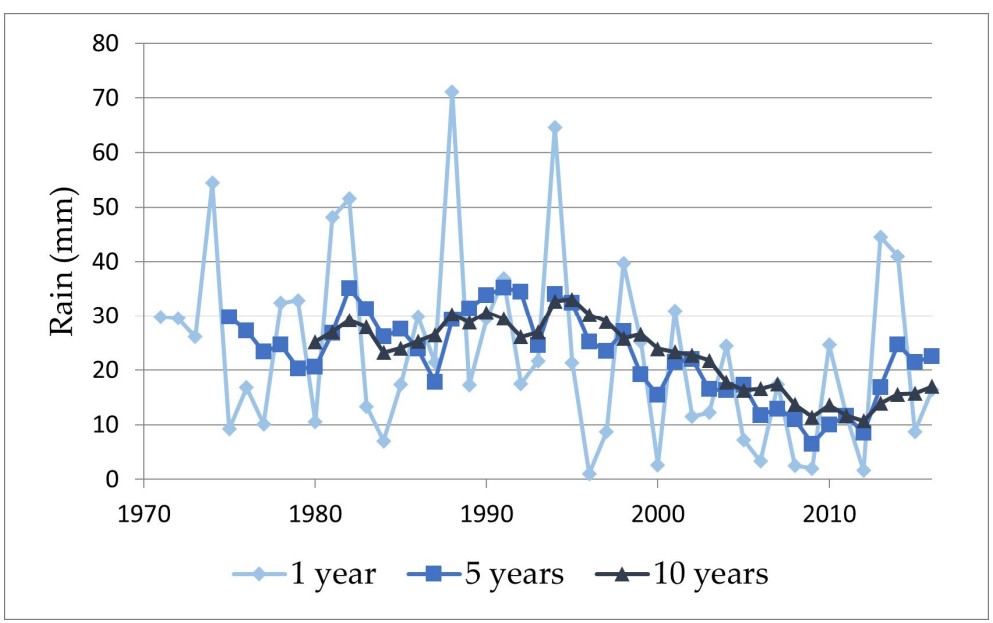

**Figure 2.** Eilat annual rainfall (mm): five-year running average, 10-year running average (Israel Meteorological Service).

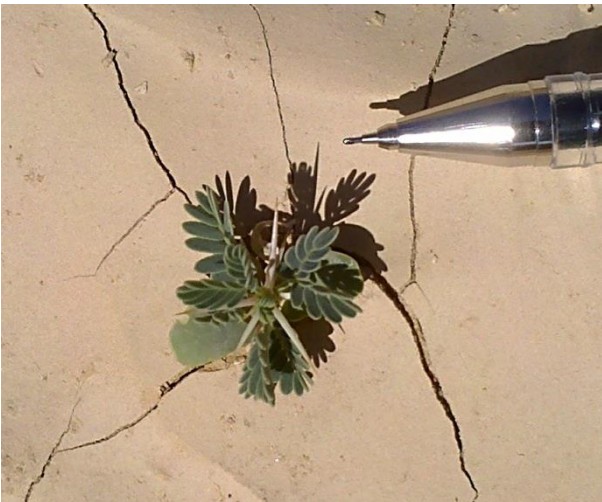

**Figure 3.** A one to two-month-old *A. Raddiana* seedling found in Wadi Shlomo.

Wadi Roded—The wadi drains an area of 5 km$^2$ before reaching the monitoring plot; the total drainage basin of Wadi Roded is about 100 km$^2$. The area starts at a vast 130 m-wide alluvial channel that encounters a magmatic rock barrier; the channel is then incised in the rock, and flow direction changes by 90 degrees from northeast to southeast and narrows to a range between 10 and 40 m (Figure 4).

Wadi Ktora—The wadi drains an area of 25 km$^2$. The section of the wadi includes a braided channel, which consists of a network of small channels separated by small islands. The main channel flows southeast. The channel is narrow in the northern part of the study area and continues to flow close to the northeastern bank with small channels that reach the central area and southwestern bank. The width of the wadi at its narrowest point (northern part) is about 35 m and widens to more than 500 m in the central area of the wadi [39].

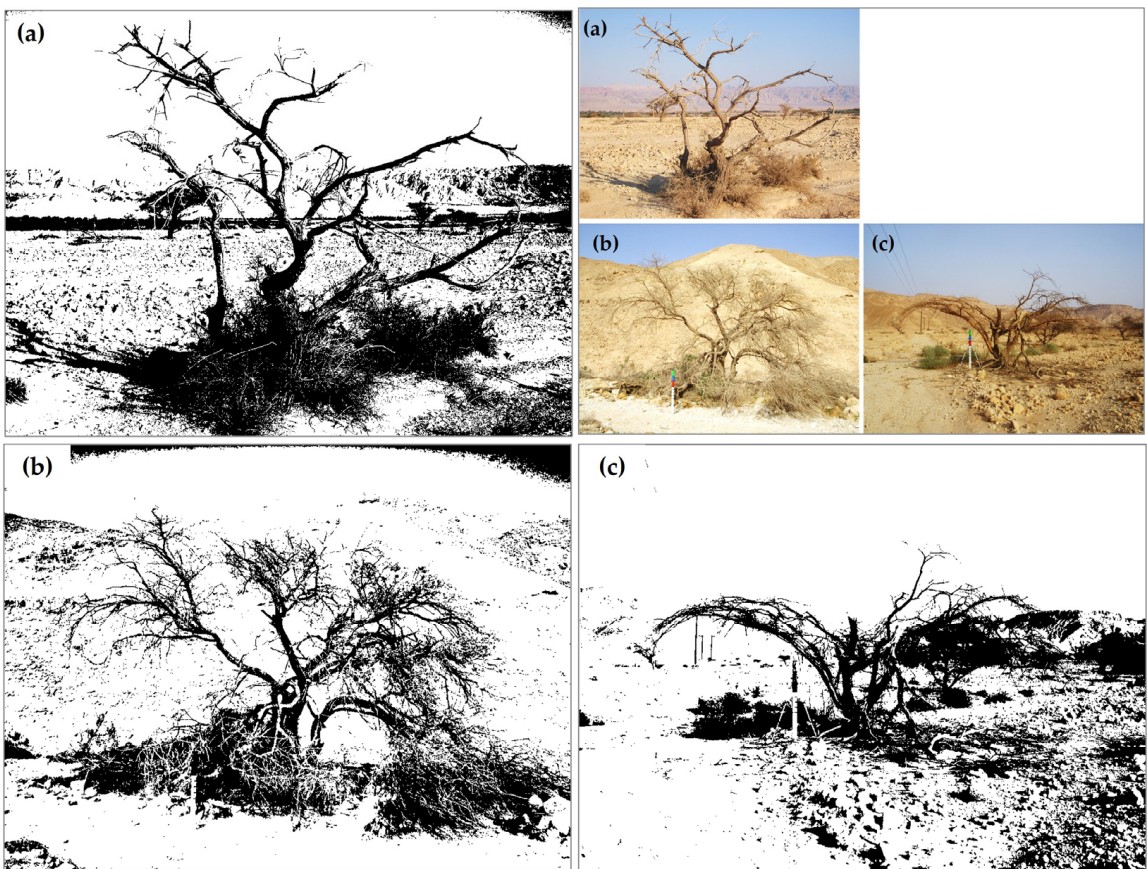

**Figure 4.** Black and white images of selected *A. raddiana* wood skeletons in Wadi Ktora in different stages of decomposition. Right-top: The original RGB images: (**a**) Year of death is unknown; >10 years. (**b**) Dead tree eight years after death. (**c**) Dead tree one year after death. The wood skeleton of a tree, one year after dying (Figure 4c), has fewer branches and appears to be in an advanced stage of decomposition compared to the wood skeleton of a tree that died seven years earlier (Figure 4b).

### 2.1.2. Margins of the Salt Flats

Yotvata and Evrona salt flats are the largest salt flats of the southern Arava Valley. These endorheic basins are the lowest part of closed systems that drain the wadis and alluvial fans from the east, west, and north [40]. The shallow water table is close to the surface; thus, water evaporation leaves high salt concentrations in the ground [41]. Vegetation surrounds the salt flats in strips according to salt tolerance [41]. The shallow water table supports a rich and dense cover of halophytic vegetation [40]. The dense acacia tree populations (*A. raddiana* and *A. tortilis*) dominate the landscape at the margins of the salt flats (Figure 5). The monitoring sites of Yotvata and Evrona are located east of the Arava Highway (Road 90).

### 2.2. Long-Term Field Monitoring

Since 2000, a continuous survey of the three acacia species in seven different sites has been conducted yearly in the southern Negev and Arava (established by B. Shalmon, ecologist of the Israel Nature and Parks Authority). This study focuses on five sites in the southern Arava Valley which host two acacia species: *A. raddiana* and *A. tortilis*. In each site, a minimum of 45 trees were marked with a numbered metal tag, and their location was identified using a GPS for revisitation of each tree every year. The individual-based monitoring data includes 16 successive years of tree trunk circumference measurements. Yearly population monitoring enabled us to determine the mortality and recruitment rate of the acacia populations as well as the growth rates of individual trees.

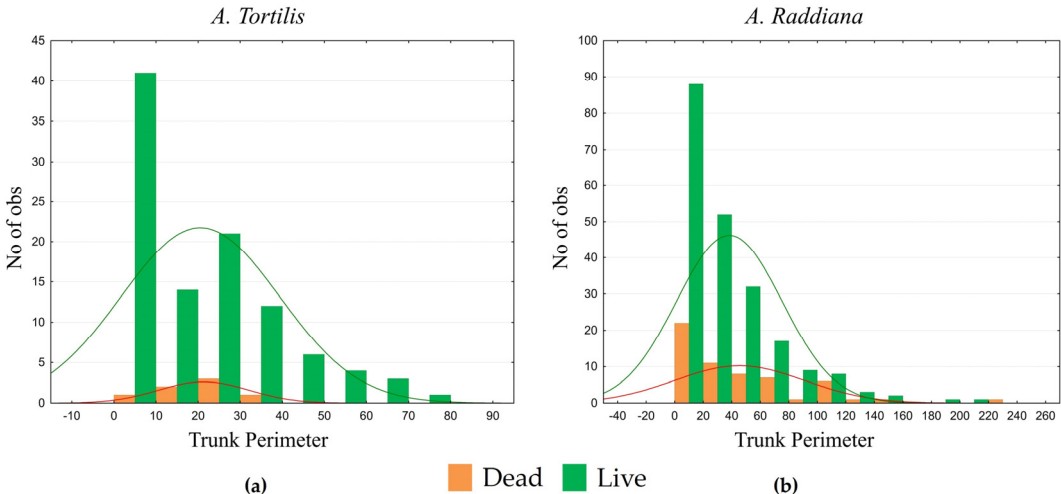

**Figure 5.** Size-frequency trunk perimeter (cm) distribution of dead and live trees: (**a**) *A. tortilis*; (**b**) *A. raddiana*.

This survey provides a unique database of the acacia population dynamics during a sequence of dry years that ended in a vast flood during the winter of 2010.

### 2.2.1. Population Dynamics: Recruitment

During each monitoring event, we scanned the entire site area looking for new seedlings. Very small seedlings were considered to be up to one year old (Figure 3).

### 2.2.2. Population Dynamics: Mortality

This study included two types of mortality estimations: the percentage of dead trees from the entire population at the site each year (2000–2016) and the absolute number of trees that died each year during the monitoring years (2001–2016). The first estimate is a snapshot measure, but the second can be used to infer mortality rates. Because dead acacias remain standing after dying, the tree mortality percentage represents the accumulative mortality of an unknown number of years. Nevertheless, most studies on acacia population dynamics use this parameter to estimate tree mortality. The yearly temporal resolution of this study enables the unique database of the annual mortality rates of acacia trees in the southern Arava.

For each dead tree, the year of dying was recorded, and a ground image was taken. The number of branches and the order of branches were analyzed manually.

To define the minimal effective rain needed for survival, we suggested using only rain events that were larger than a certain threshold number. As rainfall data resolution is monthly, not per event, we used a monthly threshold (mm) to calculate annual rainfall from 1980 to 2016. The correlation between the yearly mortality percentage and rainfall running average was tested for all monthly thresholds from 0 to 15 mm.

### 2.3. Remote Sensing Analysis

A Corona image from 1968 and an aerial photograph from 2010 were used to detect individual changes. Declassified panchromatic images of the study area from 1968 to 1969 can be download for free from the "Corona Atlas of the Middle East" website (CORONA, http://corona.cast.uark.edu/ accessed on 10 February 2023). The images have a spatial resolution of 1.8 m at the nadir. We used a Corona image acquired in August 1968, and geo-rectification was made according to Barnes [42].

We compared the image to an RGB aerial photograph from 2010 having a spatial resolution of 0.25 m. To facilitate comparison, we converted the RGB photo to a panchromatic photo using all three bands. Additionally, we degraded its spatial resolution and converted 0.25 m per pixel to 2 m per pixel.

A manually digitized change detection was done separately on the two polygonal maps; each tree on the 1968 map was manually marked as present or absent in the 2010 map and vice versa. Based on the comparison, trees were divided into three categories; "new" (appearing in 2010 but not in 1968), "old" (present in 2010 and 1968), and "missing" (appearing in 1968 but not in 2010). It should be noted that it was not possible to distinguish between live and dead acacia trees from RGB aerial photographs [39]; therefore, trees in the "new" and "old" categories may either be dead or alive. Therefore, the missing trees are only a portion of the dead trees in the field. The mortality status for each tree was documented during field monitoring, resulting in a mortality rate for each of the three categories. We intentionally termed the trees that were not found in the 2010 image as missing, not dead.

## 3. Results and Discussion

### 3.1. Long-Term Field Monitoring

3.1.1. Identification of Dead Trees

No correlation was found between the number or order of branches on dead tree skeletons and the number of years since the trees had died. All trees that died during the 15 years of monitoring had branches of the fifth-order at least. Old skeletons of trees that died before monitoring was initiated (in 2000) had a variety of morphological appearances, including trunks standing without any branches. We found it impossible to estimate the elapsed time since the tree died within a time span of 15 years by it morphology (Figure 4).

Acacia trees with trunks with a circumference greater than 5 cm that had died during this 15-year survey were still standing at six of the seven sites. (In Wadi Roded, three dead trees were uprooted and removed by flash floods.) This finding conflicts with the assumptions previously made by Peled [14] and Ward and Rohner [17] that dead acacia trees remained standing for 10 years on average. Our data showed that dead acacias remained standing for at least 15 years in hyper-arid zones. This information is significant when analyzing tree population dynamics and mortality rates from a snapshot of observed mortality percentages.

3.1.2. Mortality and Tree Size

The mean trunk circumference of the dead trees was larger than for the live trees for all species, but the difference was not statistically significant (*A. tortilis* Student's *t*-test: mean dead = 28 cm; mean live = 26.8 cm; $t = 0.21$; df = 77; $p = 0.831$) (*A. raddiana t*-test: mean dead = 51.3 cm; mean live = 42.6 cm; $t = 1.49$; df = 230; $p = 0.137$). Furthermore, the size-frequency distributions of dead vs. live trees were not significantly different according to Kolmogorov–Smirnov tests for each species ($p = 0.15$) (Figure 5). Our results corresponded with those of Wiegand et al. [19], who found no relationship between size and mortality in acacia trees in the Negev. This analysis is necessary to disprove the possibility of the natural mortality of old trees.

3.1.3. Rainfall and Mortality Response Time

The relationships between annual rainfall and tree mortality revealed different patterns when analyzed for the cumulative mortality percentage and absolute number of dead trees per year (Table 1). Only trees that died during the previous year fit best with the five-year running average rainfall. In contrast, the cumulative mortality percent fit best with the 10-year running average rainfall (see Figure 2 for annual running average).

A vast regional storm in the winter of 2010 marked the end of the drought episode (Figure 2). It was the first year since 2000, when long-term monitoring started, and significant flash floods occurred at all monitoring sites. However, tree mortality rates did not significantly decline until 2014 (Figure 6). During the four years of monitoring from 2012 to 2016, a change in the trend of the running average of rainfall is evident (Figure 2), but tree mortality at the small wadi sites remained high and departed from the trend line of the rainfall/mortality correlation (Figure 7—red circles).

**Table 1.** Mortality in the small wadi sites: Pearson coefficient r and *p*-value for regression of running average rainfall and (A) count of dead trees per year; (B) cumulative mortality percent. Bold figures: correlations are significant at *p* < 0.05. Boxed frames are the best fit correlations for each mortality parameter tested.

| Running Average | (A) 2000–2016 Number of Dead Trees per Year | | (B) 1981–2016 Cumulative Mortality Percent | |
|---|---|---|---|---|
| | r | *p*-Value | r | *p*-Value |
| 1 Yr | **−0.604** | *p* = **0.013** | −0.128 | *p* = 0.552 |
| 2 Yr | **−0.597** | *p* = **0.015** | −0.173 | *p* = 0.418 |
| 3 Yr | −0.464 | *p* = 0.070 | −0.247 | *p* = 0.245 |
| 4 Yr | **−0.731** | *p* = **0.001** | −0.363 | *p* = 0.081 |
| 5 Yr | **−0.767** | *p* = **0.001** | **−0.496** | *p* = **0.014** |
| 6 Yr | **−0.524** | *p* = **0.037** | **−0.583** | *p* = **0.003** |
| 7 Yr | **−0.586** | *p* = **0.017** | **−0.658** | *p* = **0.000** |
| 8 Yr | **−0.625** | *p* = **0.010** | **−0.762** | *p* = **0.000** |
| 9 Yr | **−0.541** | *p* = **0.030** | **−0.825** | *p* = **0.000** |
| 10 Yr | −0.421 | *p* = 0.104 | **−0.853** | *p* = **0.000** |

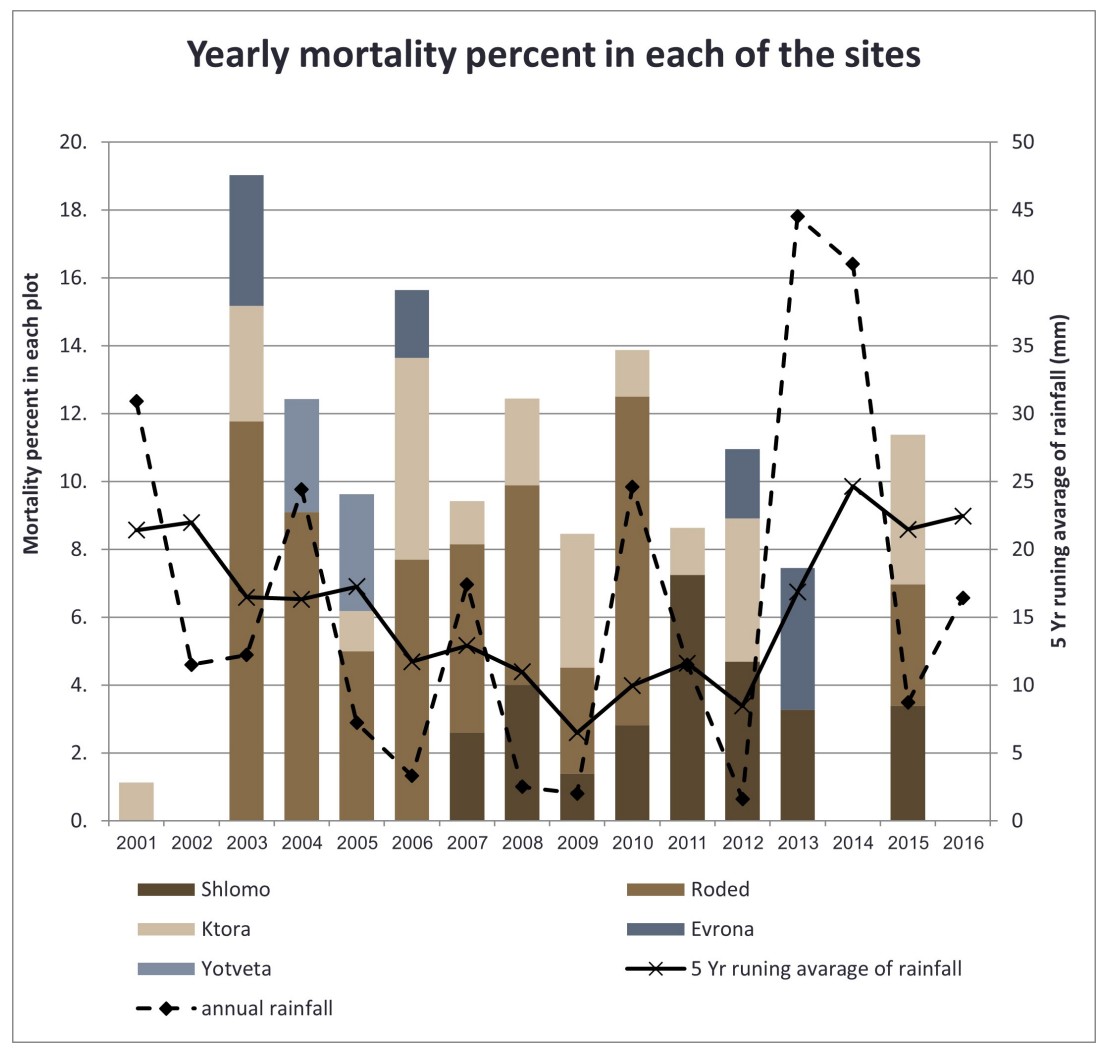

**Figure 6.** Annual mortality percentage in each site (colored columns), five-year running average rainfall (continuous line), and maximal annual monthly rainfall (dashed line). Brownish columns represent small wadis (Shlomo, Roded, Ktora). Bluish columns represent salt flats (Yotvata, Evrona).

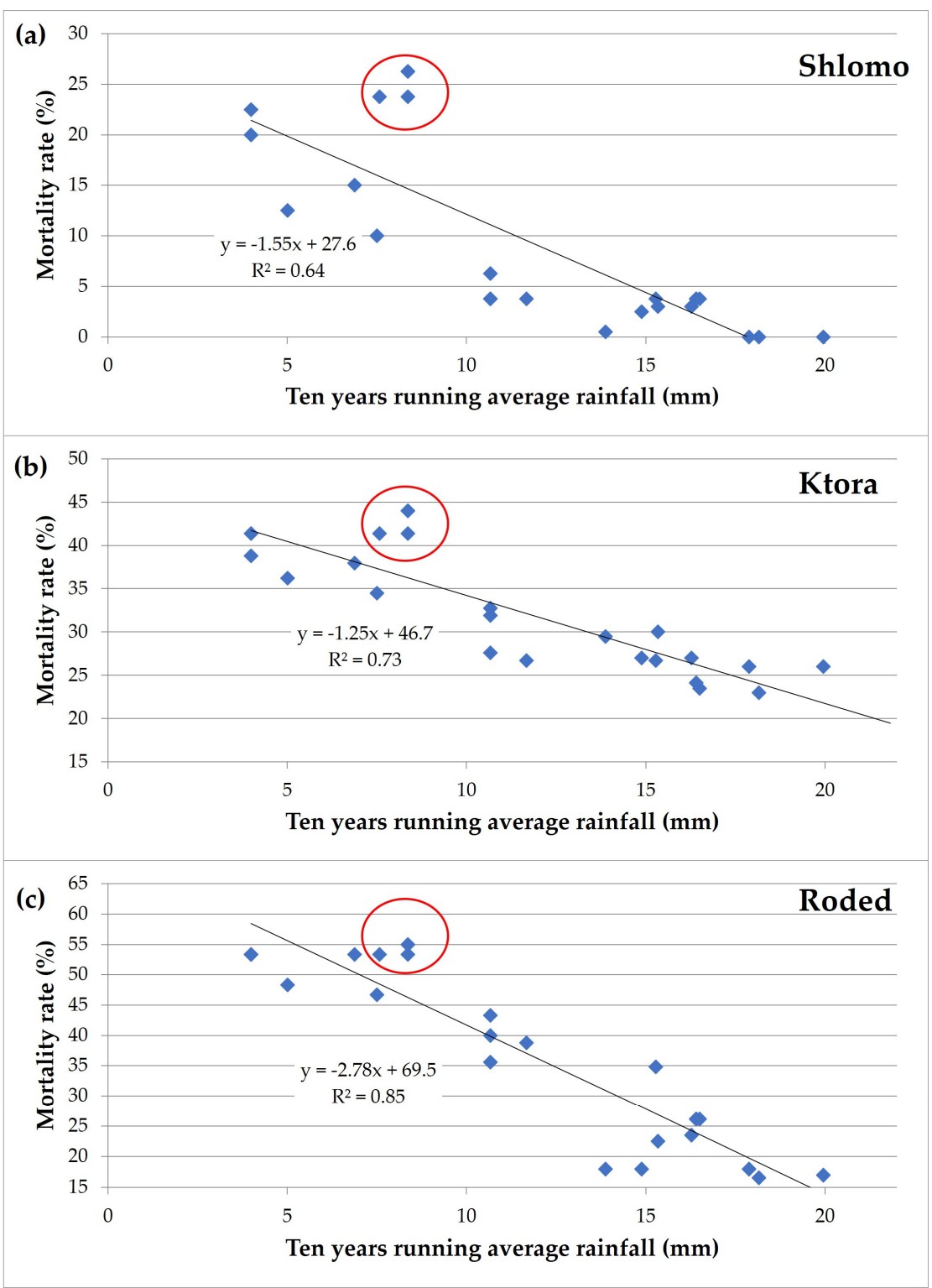

**Figure 7.** Linear regression of 10-year running average rainfall and mortality percentage in three sites: (**a**) Shlomo (r = −0.8; *p* = 0.0001), (**b**) Ktora (r = −0.85; *p* = 0.0001), (**c**) Roded (r = −0.92; *p* = 0.000). Modified by Peled [13] (1981–1987) and this study (2000–2016). The dots in the red circle represent the years 2013–2016.

The percentage of dead trees at a site was a cumulative variable that included all standing dead trees (see Section 3.1.1. Identification of Dead Trees). Therefore, it is expected that this variable will correlate weakly with the last rainy season but will correlate better

over a longer time span. In addition, the mortality process of a tree is long, usually more than one year [14]. Hence, we can expect the annual mortality to reflect the water supply of the past several years rather than only of the previous year. The recruitment rates on the other side of the population dynamics balance also depended on rainfall of more than one year. Low recruitment was usually due to unsuccessful tree establishment rather than insufficient germination. The deviation from the trend line of the mortality percentage, between 2012–2016 at the small wadi sites (Figure 7: red circles), was mainly an outcome of very low recruitment rates even after the rainfall trend shifted after 2010 (Figure 2). Thus, additional years of long-term monitoring records are essential for determining how long and how much rain is needed for population turnover and recovery.

### 3.1.4. Rain Impact at the Different Hydrological Regimes

Linear regression between the five-year running average rainfall and the yearly number of dead trees showed a strong negative correlation at the small wadi sites ($r = -0.78$; $p = 0.000$) but no correlation at the salt flat sites ($r = 0.0016$; $p = 0.99$) (Figure 8). Large wadi site regressions were not performed because of the distance between those sites and the meteorological station.

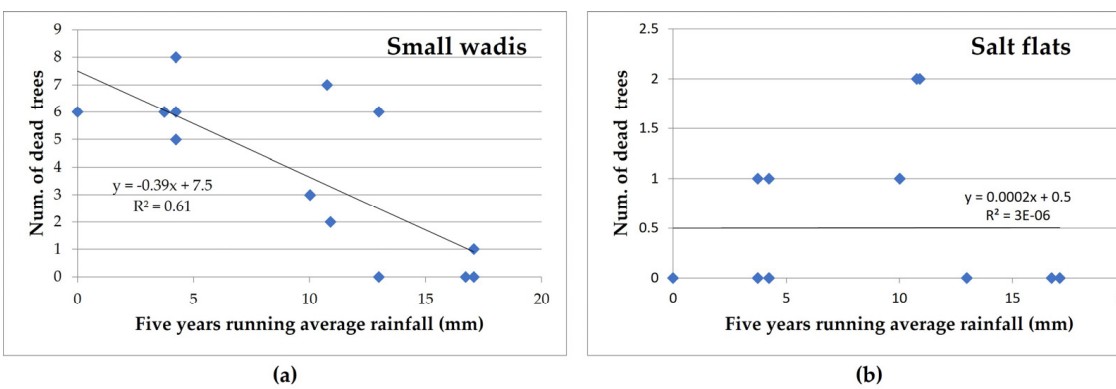

**Figure 8.** Linear regression of five-year running average rainfall and the sum of yearly dead trees at (**a**) small wadi sites: Shlomo, Roded, Ktora ($r = -0.78$; $p = 0.000$) and (**b**) margins of salt flat sites: Yotvata and Evrona ($r = 0.000$; $p > 0.05$).

While the mortality process indicated the long-term impact of water supply, the canopy status of a tree was a short-term indicator of the water supply [39]. The annual mean canopy vigor estimation at the small wadi and salt flat sites revealed a moderately positive correlation ($r = 0.6$; $p = 0.02$) and no correlation ($r = 0.3$; $p = 0.23$) with rainfall, respectively. Caution had to be taken in this comparison because the two acacia species (*A. raddiana* and *A. tortilis*) were not evenly represented at the sites. While the population at the small wadi sites was composed of only 6% *A. tortilis* and 94% *A. raddiana*, the salt flat sites' trees were composed of 64% *A. tortilis* and 36% *A. raddiana*. *A. tortilis* is known as a more thermophilic tree that can be found in drier regions than *A. raddiana*. Therefore, the response of *A. tortilis* to rainfall may be less pronounced in canopy vigor estimation than the response of *A. raddiana*. To test the validity of the comparison, we compared only *A. raddiana* trees between small wadis ($n = 217$) and salt flats ($n = 54$), and the difference between the two environments was even greater ($r = 0.62$, $p = 0.01$; $r = 0.008$, $p = 0.6$ respectively).

These results indicated that the acacia trees in the salt flats do not rely directly on rainfall, as do the trees in the small wadis. To date, comparisons between these two environments as acacia habitats have not been published.

The growth of *A. raddiana* trees was documented during the long-term field monitoring in two plots representing the two hydrological regimes: small wadi (Ktora) and salt flat (Yotvata) (Figure 9). The trees were categorized based on trunk circumference in 2000, the first year of the monitoring. At both sites, some growth was recorded in all categories

throughout the study, and the growth percentage was greater for a smaller initial tree size (Table 2). However, while all categories of the salt flat's trees exhibited consistent growth during years of the monitoring, at the small wadis, the rate was moderate and in some years decreased (Figure 9).

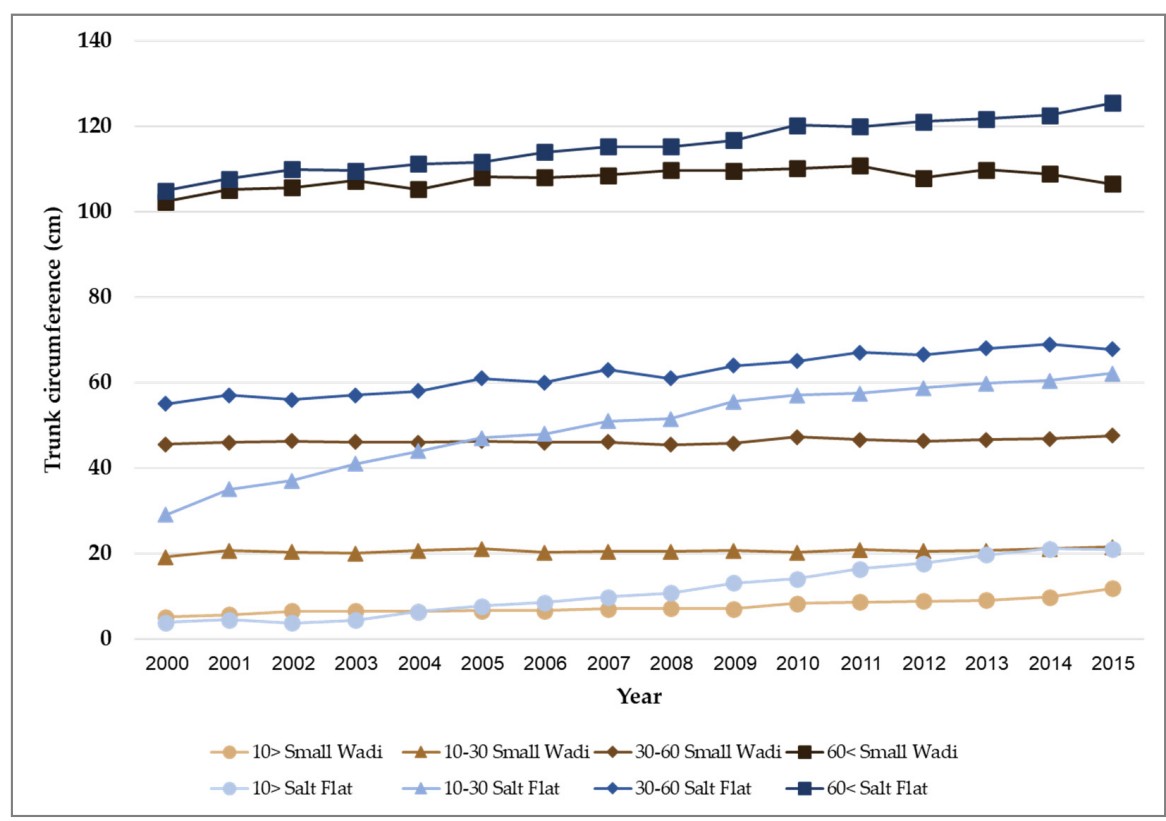

**Figure 9.** Mean trunk circumference of *A. raddiana* trees, grouped by size, for the years 2000–2015 in two sites: Wadi Ktora (small wadi) and Yotvata (salt flat), representing different hydrological regimes.

**Table 2.** Growth increment percent of 15 years.

| | Growth Percent (2000–2015) | |
| --- | --- | --- |
| Trunk Circumferences Size Category (cm): | Small Wadi | Salt Flat |
| <10 | 131 | 441 |
| 10–30 | 12 | 114 |
| 30–60 | 4.4 | 23.3 |
| >60 | 4 | 19.6 |

3.1.5. Effective Rain for Trees Survival

At all sites, the strongest correlation was found to be with a rainfall threshold of 8–10 mm. Months with lesser rainfall were excluded from the total annual rainfall. (Figure 10). The correlation was strongest in the Roded site and did not change much with or without any threshold up to 9 mm. For Shlomo and Ktora, the highest correlations were found to be with thresholds of 9 and 10 mm, respectively. Those results may suggest that rain events lower than 8 mm did not increase acacia trees survival rates. Differences in the effective rainfall (threshold found) between sites might have resulted from the variance between drainage basins' attributes. However, variations of the effective rain among the three sites were not significant (1–2 mm).

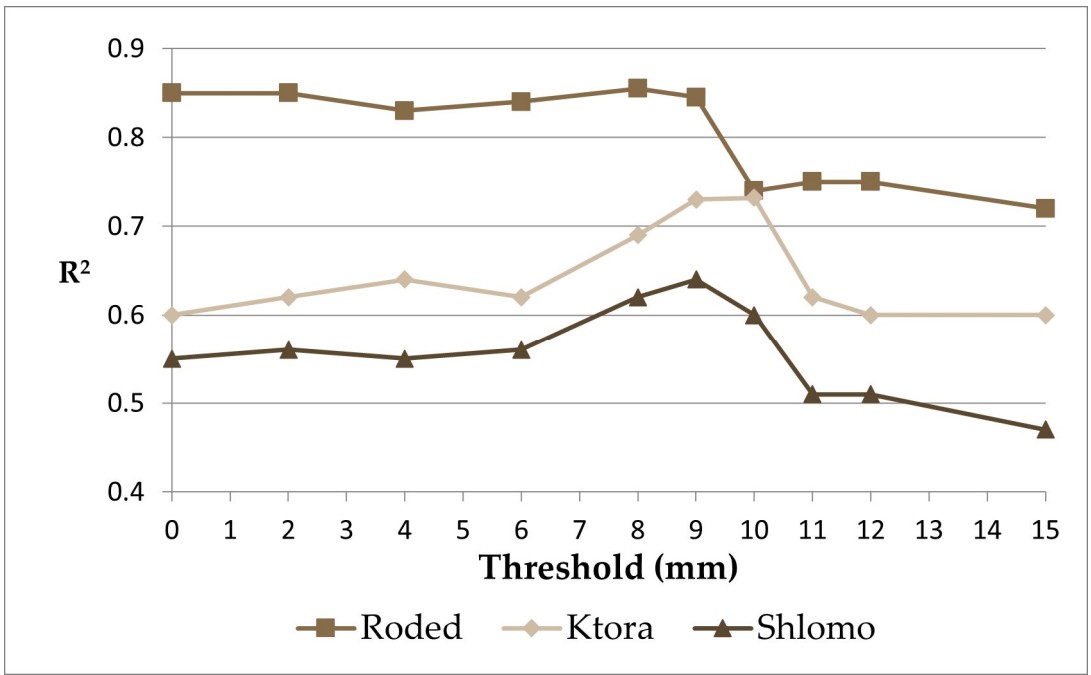

**Figure 10.** $R^2$ values for different rainfall thresholds (mm) of the regression of 10-year running average rainfall and mortality percent at small wadi sites: Shlomo, Roded, Ktora.

### 3.1.6. Effective Rain for Recruitment

During 2000–2012, no new seedlings were found at any of the sites, including after the vast flood events of January 2010. In 2012, after a considerably high rainfall was recorded in October–November (25.8 mm, Figure 2), new seedlings germinated: Shlomo (14), Roded (4), Ktora (1). More seedlings germinated and were found in 2014 at Roded (2) and Ktora (8) (Table 3). Most seedlings (79%) died during their first year and only 6% at all sites survived for more than four years and were still alive in 2016.

**Table 3.** Summary of new seedlings found in the monitoring sites 2001–2016.

| Site | Coordinates | Estimated Year of Germination | Number of Seedlings Found | Seedlings Survived in 2016 | Years of Seedlings Dead | Survivability (%) |
|---|---|---|---|---|---|---|
| Roded | 29°36′43 N/34°54′50 E | 2012 | 4 | 2 | 2013 | |
| Roded | | 2014 | 2 | 0 | 2015 | 33.3 |
| Shlomo | 29°32′56 N/34°54′00 E | 2012 | 14 | 1 | 2013 | 7.1 |
| Ktora | | 2012 | 1 | 0 | 2014 | |
| Ktora | 30°01′15 N/35°04′50 E | 2014 | 8 | 1 | 2015 | 11.1 |

A crucial factor for tree population longevity is the survival of seedlings in their first year. While the seed is the phase of highest resistance to extreme environmental conditions in the plant life cycle, the seedling phase is the time when the plant is most vulnerable to environmental conditions [43]. The low percentage of seedling survival was is in agreement with Stavi et al. [44] who found 2.5% survival in the Southern Arava, ranging from 0 to 23% at nine different wadis. The survey each year was carried out during November–December. We assumed that more seedlings germinated after the winter floods, but only a few had survived the dry summer conditions of their first year. Therefore, the actual percentage of surviving seedlings was probably lower than our records showed. The establishment of the seedlings was the main limiting factor in the recruitment of the acacia population, not germination [10].

The lack of new seedlings after the regional storm of the winter of 2010 reinforces the assumption [29] that winter rain and flood events are ineffective for germinating and establishing new acacia seedlings. This is due to low and unfavorable germination temperatures and the relatively short time available for a seedling to develop a root system that will enable it to survive the dry summer. Halevy and Orshan [29], however, suggested that autumn floods are essential for germination and survival of new acacia seedlings.

### *3.2. Extending the Temporal Scope of Mortality Monitoring Using Satellite Images*
#### 3.2.1. Population Dynamics

Despite a high mortality percentage at most sites (Figure 6), no clear evidence of population decline was revealed when the monitoring time scale was extended using historical satellite images (Figure 11). Comparing regeneration and mortality rates from 1968 and 2000, the number of new trees exceeded dead trees at all the sites. However, considering the sequence of drought years that ended in 2010, an additional period was assessed: from 1968 to 2010. The results imply that between 2000 and 2010, tree populations at all sites experienced less renewal and more death than in the preceding decades. Two sites, Roded and Ktora, suffered a turnover of the regeneration–mortality ratio, i.e., total population decline. However, this period of high mortality may be episodic and not indicate a constant trend.

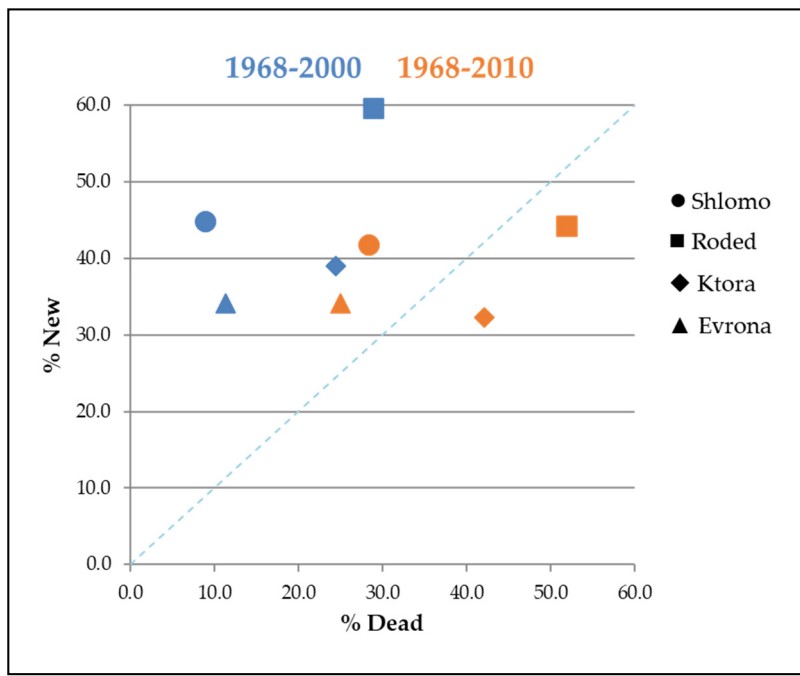

**Figure 11.** Acacia population dynamics (regeneration vs. mortality rate) by site for two periods: 1968–2000 and 1968–2010.

#### 3.2.2. Mortality Time Scale Episodes

The extracted historic status of each tree revealed the long-term population dynamics in each plot and the variations among them (Figure 12). The overall proportion of old and new trees was near parity at the Ktora (49% old; 51% new) and Shlomo (51% old; 49% new) plots. In contrast, Evrona had more old trees (63%) than new, while the most prominent difference was recorded for the Roded plot, where most trees were new (70%). When considering only living trees, the majority of new trees in the Roded plot was even greater as old trees counted for only 8%. The ratio between old and new trees implied long-term processes affecting population dynamics. Because the mean age of the acacia trees in this area is unknown, it was difficult to predict the expected ratio between those at least 50 years old and younger trees.

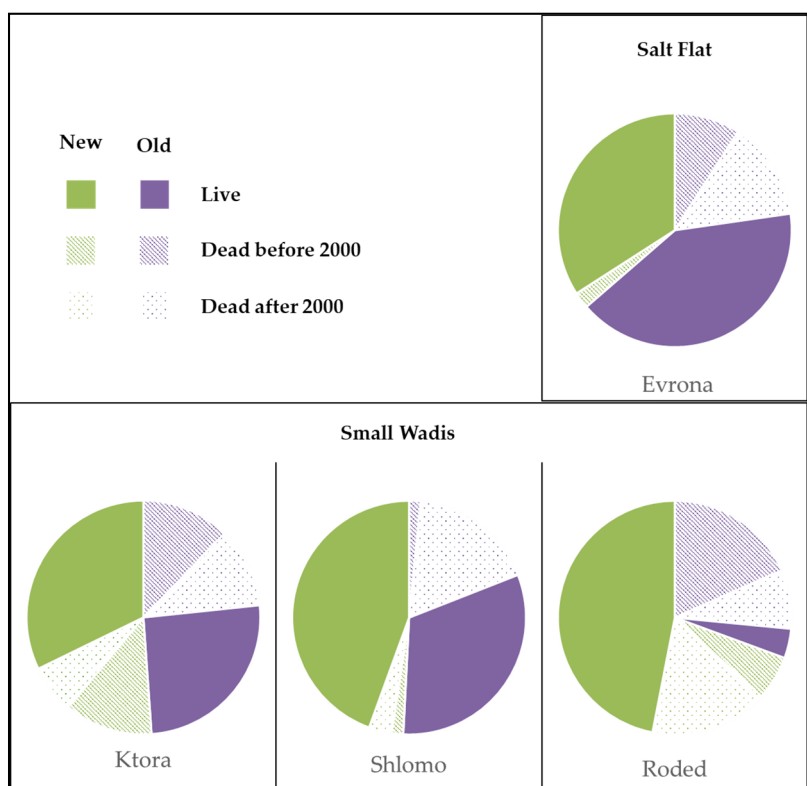

**Figure 12.** Population mortality analysis according to tree status and timing of death in each plot.

The Ktora and Roded plots had the highest mortality ratio: 42 and 49%, respectively (Figure 11); furthermore, the portion of new versus old dead trees in those plots was considerably higher compared to those of Shlomo and Evrona, where only few of the dead trees are new. The mortality of the Shlomo populations was evidently recent (Figure 12); 87% of the dead trees died after 2000. Likewise, the first record of a dead tree in this plot was from 2007, while the other plots ranged from 2001 to 2003 (Figure 6). In the Evrona and Shlomo populations, the vast majority of the dead trees were "old" (Figure 1), i.e., detected in the 1968 image. This may have implied that those populations were more stable and subject to a more moderate process of decline. Overall, each plot showed a unique pattern of mortality dynamics and no pattern was common to the small wadi plots versus the salt flats.

## 4. Conclusions

To assess recruitment, mortality, and survivorship rates, long-term data must be collected, especially for species like acacia trees, which have a lifespan of hundreds of years. This study demonstrated an ideal integration between detailed but time-limited field monitoring data and the use of historical satellite images to expand the monitoring timeframe. Moreover, this approach helped identify and explain spatial and temporal variances between the tree populations, thereby avoiding a biased interpretation of mortality observations.

These results indicated that the annual mortality of acacia trees in small wadis reflected the cumulative effective rain events in the preceding five years, whereas the acacia population on the salt flats were not affected by annual rainfall fluctuations. This is the first publication comparing acacia trees in these two environments. The mortality pattern and dynamics of each plot was unique, suggesting unsynchronized mortality and recruitment episodes on a regional scale. By extending the temporal scope of monitoring tree population dynamics as applied in this study, we gained new insight into the long-term impacts of climate fluctuations on ecosystems and explained current the population status from a historic perspective.

**Author Contributions:** Conceptualization, S.M., J.E.E., S.R. and D.G.B.; methodology, S.I.; software, S.I.; validation, S.M., J.E.E., S.R. and D.G.B.; formal analysis, S.I.; investigation, S.I.; resources, B.S. and S.I.; data curation, B.S. and S.I.; writing—original draft preparation, S.I.; writing—review and editing, S.I., J.E.E., O.K. and S.M.; visualization, S.I.; supervision, J.E.E., S.R. and D.G.B.; project administration, S.M.; funding acquisition, S.M., J.E.E., S.R. and D.G.B. All authors have read and agreed to the published version of the manuscript.

**Funding:** This research was funded by the Israel Ministry of Science and Technology. S.I. was supported by the Negev-Zin Scholarship and the Yair Guron Memorial Scholarship.

**Data Availability Statement:** Data sharing not applicable.

**Acknowledgments:** We would like to thank all the supporting organizations of the monitoring: Israel parks authority, JNF, Hamaarag, and the acacia Research Center. We would like to thank the many workers and volunteers of the monitoring.

**Conflicts of Interest:** The authors declare no conflict of interest.

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
