# Peer review of "Long-Term Monitoring of Tree Population Dynamics in Desert Ecosystems: Integrating Field and Satellite Data"

_land, doi:10.3390/land12081640_

Round 1

Reviewer 1 Report

Please find the comments attached in the pdf file.

Please review the verb tenses in the different sections. Results should be expressed in past tense.

Author Response

Following are the changes we made which correspond to the comments made by the reviewer:

  1. Acacia

We corrected all places where 'acacia' appeared with capital letter.

  1. Comments regarding the citation style:

We changed the style as instructed, and deleted years (line75,206).

  1. Please rephrase this conclusion (Line 30-32)

Lines 31-33: We rephrase this sentence:

Combining field monitoring and historical satellite image data provides a unique database of the response of acacia population dynamics. This unique records reveals the response of acacia population to climate fluctuations and shows a period of episodic mortality.

  1. Please do not repeat keywords from the title

Lines 34-35: We deleted: long term monitoring; population dynamic;

  1. richer?? (Line 40)

Line 41: We accept the suggestion.

  1. Grammer mistake – Line 93: corrected.
  2. Spelling (Line 92)- Line 94:
  3. I would rephrase the objective of the work to make it more explicit (Line 93-95)

Line 95-97: We added the objective of the paper:

The objective of this study is to describe how a decrease in annual precipitation affected acacia tree population dynamics in two hydrological regime types: small wadis and salt flats.

  1. Please, it would be necessary to include a general situation map (figure 1).

Figure 1: We edited the map.

  1. A halophily index of the flora composing each study locality could be taken into account according to the ehaloph classification. https://ehaloph.uc.pt/listplants (Line 151).

We respectfully disagree with this comment. While we acknowledge the importance of understanding salinity stress in arid environments, we do not see the relevance of a community-level halophily index for the long-term demography of this tree genus. In our opinion, such information does not strengthen or weaken any of our observations, arguments and conclusions. This is not explained anywhere in M&M section. Please, explain how this calculation is performed and how the non-correlation result is achieved. (Line 209).

Line 190-191: We added the following to the M&M section:

For each dead tree, the year of dying was recorded, and a ground image was taken. The number of branches and the order of branches were analyzed.

  1. How is this taken into account when analyzing remote sensing data? (Line 221-222)

See Remote sensing analysis section, lines 215-217:

It should be noted that it is not possible to distinguish between live and dead acacia trees from RGB aerial photographs [39]; therefore, trees in the "new" and "old" categories may either be dead or alive.

  1. If there are no significant differences we cannot assume that there is overall differentiation between the groups. Have the data been checked for normality? If the data are not normal, nonparametric statistics should be used. (Line 232).

We agree with this claim, dead trees and live trees apparently do not differ from each other by size parameters. The data was found with normal distribution by Kolmogorov–Smirnov test.

  1. Grammer (Line 269) – Line 283 -Corrected
  2. This paragraph should be placed in Methods section (Line 335-340)

Lines 193-199: We removed the paragraph as suggested.

  1. Please review the verb tenses in the different sections. Results should be expressed in past tense.

We corrected all verb tenses.

Reviewer 2 Report

This is an interesting paper, but you may improve this article in order to publish in this journal. Otherwise, I have a lot of recommendations to increase the quality of your manuscript. Be careful with the writing and mistakes.

There is a keyword repeated in the article title. The keyword is “long term monitoring”. In order to increase the visibility of your paper I recommend changing this keyword. If you change it by other keyword, you will increase the probability that your paper could be found by future readers when they look for your paper in some databases like Scopus for example. If you repeat the same words in the article title and in keywords, less people could find your work. So, you must think about the visibility of your research.

As well you must put in alphabetical order all the keywords.

In the whole paper when you write “Acacia” in italics is wrong written because you must write it in italics. If you write “Acacia” in capitals that means that is a scientific name, so you must write it in italics and the very first time you must write its authors. If you write “acacia” with no capital letters is perfectly written.

Line 37. You must write the authors of Vachellia.

Line 38. You must follow the rules of this journal. You must write “[1,2]” instead of “[1], [2]”. This is a very common mistake that you must fix in order to publish in this journal. You must download a paper of this journal and copy its writing style.

Line 42. You must follow the rules of this journal. You must write “[5,6]” instead of “[5], [6]”. This is a very common mistake that you must fix in order to publish in this journal. You must download a paper of this journal and copy its writing style. Please, fix this mistake in the whole manuscript.

Line 54. You must follow the rules of this journal. You must write “[8,11–19]” instead of “[8], [11] –[19]”. This is a very common mistake that you must fix in order to publish in this journal. You must download a paper of this journal and copy its writing style. Please, fix this mistake in the whole manuscript.

Line 79. You must write in italics the scientific names and its authors the very first time.

Line 91. Just after the word “Nevertheless” you must write a comma, not a point. Please, fix this mistake.

Line 110. You must write the coordinates of each point in the table 3.

Line 114. You must improve the map. It is very difficult to see the countries. As well you must write the wadis.

Line 146. You must write in the map the localities of this line.

Line 173. You must write the name of the species of this photograph because a Figure must explain itself.

Line 190. There is no need to write the year of the reference in this journal.

Line 215. Please, fix this mistake. You must fix the text “(Error! Reference source not found)”. This is a very common mistake in the whole paper.

You must write the Conclusions section.

All the references do not follow the rules of this journal. Please, download a paper and copy its style. You must to avoid the word “vol.”, “no.” and “pp.”. Please, follow the instructions of this journal. As well some years of the references are wrong located.

Otherwise, the authors adequately developed the Introduction, presenting the problems but you must write explicitly the objectives of this paper.

The authors are to be congratulated for the results obtained in this article.

And do not forget to write the Conclusions of this article.

Your English is good.

Author Response

Following are the changes we made which correspond to the comments made by the reviewer:

  1. There is a keyword repeated in the article title. The keyword is “long term monitoring”. In order to increase the visibility of your paper I recommend changing this keyword. If you change it by other keyword, you will increase the probability that your paper could be found by future readers when they look for your paper in some databases like Scopus for example. If you repeat the same words in the article title and in keywords, less people could find your work. So, you must think about the visibility of your research.

Lines 34-35: We exclude the word from the keywords because it is already in the title.

  1. As well you must put in alphabetical order all the keywords.

Lines 34-35: Corrected

  1. In the whole paper when you write “Acacia” in italics is wrong written because you must write it in italics. If you write “Acacia” in capitals that means that is a scientific name, so you must write it in italics and the very first time you must write its authors. If you write “acacia” with no capital letters is perfectly written.

We replaced all Acacia into acacia.

  1. Line 37. You must write the authors of Vachellia.

Line 38: We added: Maslin, B. R., J. T. Miller, and D. S. Seigler. "Overview of the generic status of Acacia (Leguminosae: Mimosoideae)." Australian Systematic Botany 16.1 (2003): 1-18.

  1. Line 38. You must follow the rules of this journal. You must write “[1,2]” instead of “[1], [2]”. This is a very common mistake that you must fix in order to publish in this journal. You must download a paper of this journal and copy its writing style.

Line 42. You must follow the rules of this journal. You must write “[5,6]” instead of “[5], [6]”. This is a very common mistake that you must fix in order to publish in this journal. You must download a paper of this journal and copy its writing style. Please, fix this mistake in the whole manuscript.

Line 54. You must follow the rules of this journal. You must write “[8,11–19]” instead of “[8], [11] –[19]”. This is a very common mistake that you must fix in order to publish in this journal. You must download a paper of this journal and copy its writing style. Please, fix this mistake in the whole manuscript.

We changed the style as instructed.

  1. Line 79. You must write in italics the scientific names and its authors the very first time.

Line 81: Corrected

  1. Line 91. Just after the word “Nevertheless” you must write a comma, not a point. Please, fix this mistake.

Line 92: Corrected

  1. Line 110. You must write the coordinates of each point in the table 3.

Table 3: We added the coordinates.

  1. Line 114. You must improve the map. It is very difficult to see the countries. As well you must write the wadis.

Figure 1: We follow your suggestions and improve the figure.

  1. Line 146. You must write in the map the localities of this line.

Figure 1: We added all study sites labels to the map

  1. Line 173. You must write the name of the species of this photograph because a Figure must explain itself.

Figure 3: We added A. Raddiana

  1. Line 190. There is no need to write the year of the reference in this journal.

Line 206: Corrected

  1. Line 215. Please, fix this mistake. You must fix the text “(Error! Reference source not found)”. This is a very common mistake in the whole paper.

Corrected

  1. You must write the Conclusions section.

See section 4

  1. All the references do not follow the rules of this journal. Please, download a paper and copy its style. You must to avoid the word “vol.”, “no.” and “pp.”. Please, follow the instructions of this journal. As well some years of the references are wrong located.

Corrected

  1. Otherwise, the authors adequately developed the Introduction, presenting the problems but you must write explicitly the objectives of this paper.

Lines 95-97: We added the objective of the paper:

The objective of this study is to describe how a decrease in annual precipitation affected acacia tree population dynamics in two hydrological regime types: small wadis and salt flats.